# Symptoms of common mental disorders and suicidality among sexually diverse men who have sex with men in Ghana

Melissa A. Stockton[1]*, Sangchoon Jeon[2], Samuel Akyirem[2], Edem Zigah[3,4]
Nii Dromo Wallace-Atiapah[4], Gamji Rabiu Abu-Ba'are[3], Richard Panix Amo-Oto[4],
Irene Ofori[5], Michael Adu[4], Kwasi Torpey[6], Laura Nyblade[7], LaRon E. Nelson[2,8]

1 Gillings School of Global Public Health, University of North Carolina at Chapel Hill, Chapel Hill, North Carolina, United States of America, 2 School of Nursing, Yale University, New Haven, Connecticut, United States of America, 3 School of Nursing, University of Rochester, Rochester, New York, United States of America, 4 Priorities on Rights & Sexual Health, Accra, Ghana, 5 Youth Alliance for Health & Rights, Kumasi, Ghana, 6 Department of Population, Family & Reproductive Health, School of Public Health, University of Ghana, Legon-Accra, Ghana, 7 RTI International, Washington, Columbia, United States of America, 8 Yale Institute for Global Health, School of Public Health, New Haven, Connecticut, United States of America

* mastockt@email.unc.edu

## Abstract

In Ghana, as in many countries globally, sexually diverse, gay and bisexual men who have sex with men (MSM) face a syndemic of HIV and mental illness. This global syndemic of HIV and poor mental health among MSM is driven by social and structural processes, including stigma, human rights abuses, and interpersonal violence that exacerbate marginalization, poverty, and illness and hamper access to health services. Many countries like Ghana lack prevalence data documenting the burden of mental health challenges among MSM. This cross-sectional study followed a waitlist-controlled randomized control trial (RCT) of a multilevel intersectional (HIV, sexual, and gender nonconformity) stigma-reduction intervention for healthcare facility staff and MSM to improve HIV testing in Accra and Kumasi, Ghana. Participants were screened for depression, anxiety and PTSD with the Patient Health Questionaire-9, General Anxiety Disorder-7, and Primary Care Post Traumatic Stress Disorder Screen for DSM-5 (PC-PTSD-5). Univariate analyses were conducted to assess the prevalence of symptoms of depression, suicidality, anxiety, and PTSD by HIV status. 186 participants completed the mental health survey. Among the sample, high prevalence of mental health concerns was reported: 52.1% mild to severe depression; 35.0% suicidality; 50.5% mild to severe anxiety; and 29.8% PTSD. The prevalence was higher among those who were living with HIV compared to HIV negative: moderate to severe depression 25.7 vs. 20.5%; suicidality 45.7% vs. 36.1%; moderate to severe anxiety 9.5% vs. 14.3%; and PTSD 50.0% vs. 30.8%. Investigation of risk and protective factors and investment in mental healthcare for MSM is needed.

**Data availability statement:** Our data set is not available for sharing publicly because we do not have ethical approval to share beyond our immediate research team. The Research Ethics Boards who have imposed this restriction are: Yale University (HRPP@yale.edu), University of Ghana Noguchi Memorial Institute for Medical Research (nirb@noguchi.ug.edu.gh), Ghana Health Services (director@noguchi.mimcom.org), and University of Toronto (ethics.review@utoronto.ca) Institutional Review Boards. Deidentified data may be made available upon reasonable request and approval from the aforementioned ethics bodies.

**Funding:** This study was funded by the National Institute of Nursing Research (R01NR019009 awarded to LEN & LN). This content is solely the responsibility of the authors and does not necessarily represent the official views of the National Institutes of Health. The funders had no role in study design, data collection and analysis, decision to publish, or preparation of the manuscript. Additionally, MAS was supported by the National Institute of Mental Health under K01MH130226.

**Competing interests:** The authors have declared that no competing interests exist.

## Introduction

Sexually diverse, gay and bisexual men who have sex with men (MSM) face co-occurring and synergistic epidemics of HIV and mental health problems across the globe [1]. This global syndemic of HIV and poor mental health among MSM is driven by their overexposure to a host of inequitable multi-level social and structural processes, namely stigma, discrimination, human rights abuses, and interpersonal violence that cumulatively exacerbate marginalization, poverty, and illness and restrict access to necessary health services [2–4]. A small, growing body of research out of Africa is beginning to document these interlocking systems and their impact on HIV and mental health [5–8]; however, many African countries, including Ghana, lack even prevalence data. In Ghana, 18% of MSM are estimated to be living with HIV, with an estimated HIV prevalence as high as 42% among MSM in Greater Accra [4]. Further, Ghanaian MSM face severe stigma and discrimination, that remains codified in law, further driving inequity in health, well-being, socio-emotional development, and economic progress [9–12]. Yet, protecting the mental health of this vulnerable population has received little attention. As such, this short report aims to document the prevalence of symptoms for depression, suicidality, anxiety, experienced trauma, and post-traumatic stress disorder (PTSD) among a sample MSM in Ghana.

## Materials and methods

This study followed a waitlist-controlled randomized control trial (RCT) of a multilevel intersectional (HIV, sexual, and gender nonconformity) stigma-reduction intervention for healthcare facility staff and MSM to improve HIV testing Ghana. [12] While the RCT focused specifically on stigma and HIV outcomes, the formative work as well as engagement in the intervention revealed that mental health was a prime concern for MSM. As such, we were able to leverage the completion of the RCT to conduct this small, cross-sectional survey to document the prevalence of mental health concerns. In this short report, we briefly describe the intervention that the mental health survey participants received in the study population section. We then present the methods for the cross-sectional mental health survey.

### Purpose

This study aimed to measure symptoms of depression, suicidality, anxiety, and PTSD among MSM in Ghana. We hypothesized that there would be a high prevalence of these mental health concerns among this population.

### Study population

Participants included in this cross-sectional survey completed two stigma-reduction interventions: interpersonal-level *Nyansapo,* a Ghanian adaptation of the Many Men, Many voices (3MV) intervention [12,13], and individual-level *HIV Empathy Education & Empowerment (HIVE³)* [14]. *Nyansapo* consisted of peer, group sessions of brainstorming, role playing, small group and paired activities, and action planning on the following topics: 1) Ghanaian MSM and dual identity; 2) HIV/STD prevention

for Ghanaian MSM; 3) HIV/STD risk assessment and prevention options; 4) Intentions to act and capacity for change; 5) Relationship issues: Partner selection, communication and negotiation of roles; 6) Social support and problem solving to maintain change; 7) Building bridges and community [12]. *HIV*³ consisted of individual, one-on-one phone based text or voice contacts, in which MSM study participants reached out to trained peer mentored reached out to MSM participants for informal discussions on affirming identities, information regarding HIV prevention and testing and overring emotional support [12]. The intervention group received the intervention between August 2021 and January 2022 and the control group received the intervention between August 2022 and December 2022.

Participants eligible for the mental survey included adult MSM (ages 18 or older) who participated in the RCT [12]. All participants lived in either the Kumasi or Accra metropolitan areas, were assigned male sex at birth, and reported sexual activity with another man at least once within the previous six months of enrollment into the trial. All participants had completed all intervention activities at the time of the survey.

## Mental health and sociodemographic measures

**Patient Health Questionaire-9 (PHQ-9).** The PHQ-9 includes nine questions that assesses the presence and frequency of the nine Diagnostic and Statistical Manual of Mental Disorders (DSM) symptoms of major depression. A total score of 5–9, 10–14, and ≥15 are considered indicative of mild, moderate, and severe depression, respectively [15]. Any endorsement of question 9, which probes on thoughts of being better off dead or of hurting oneself, is indicative of suicidality. In this sample, the PHQ-9 demonstrated good reliability (Cronbach's α = 0.82).

**Generalized Anxiety Questionnaire-7(GAD-7).** The GAD-7 includes seven questions that assesses the presence and frequency of the DSM symptoms of general anxiety disorder [16]. A total GAD-7 score of 5–9, 10–14, and ≥15 are considered indicative of mild, moderate, and severe anxiety, respectively. In this sample, the GAD-7 demonstrated good reliability (Cronbach's α = 0.82).

**Primary Care Post Traumatic Stress Disorder screen for DSM-5 (PC-PTSD-5).** The PC-PTSD-5 includes six questions; an opening question that asks about prio traumatic experiences followed by a five questions that assess the core DSM-5 symptoms of PTSD asked only of those who report trauma [17]. Total scores of ≥ 4 are considered indicative of PTSD. In this sample, the PC-PTSD-5 demonstrated good reliability (Cronbach's α = 0.88).

**Sociodemographic data.** Participants provided self-reported sociodemographic information (age, education, residency, gender identity, sexual orientation), sexual behavior, and HIV status on the baseline survey at enrollment into the RCT.

## Data collection procedures

Data collection took place between 5 November 2022 and 3 January 2023, after all participants had received the intervention. Research assistants from the MSM-serving organizations that had managed data collection during the RCT reached out to the intervention participants by phone to invite them to participate in the mental health survey. In Kumasi, research assistants organized set times at different public locations for data collection, whereas in Accra research assistants met with participants individually at their homes or at the community partner's office. Participants self-administered the survey in English on tablets using the password protected CommCare application. Research assistants, fluent in both English and the local languages (e.g., Twi, Ga) were available to clarify any questions participants had about the survey items. Participants' responses could be linked to their RCT survey data via a unique alphanumeric study identification code. Research assistants were blinded to the participants responses to the surveys. All participants received referral information on where to access mental health counseling services.

## Analysis

Univariate analyses were conducted to assess the prevalence of symptoms of depression, suicidality, anxiety, and PTSD by HIV status.

Missing imputation was performed for PHQ-9, GAD-7, and PC-PTSD-5 scores separately. For participants missing ≤3 PHQ-9 items, ≤2 GAD-7 items, ≤1 PC-PTSD-5 items, the missing items were imputed using Monte Carlo Markov Chain (MCMC) approach (assuming Missing at Random) based on the item mean and covariance between items.

### Ethics

The Yale University, University of Ghana Noguchi Memorial Institute, Ghana Health Services, and University of Toronto Institutional Review Boards approved this study. All participants provided written informed consent. Study participants received 200 Ghanaian Cedis (19 US dollar equivalent) for their participation.

## Results

### Participant characteristics

186 participants of the parent RCT completed the mental health survey (Table 1). The average age was 25.2 years old (range 18–45). Approximately half of participants (n = 94) resided in Accra, while the remaining 48.2% (n = 91) resided in Kumasi. A large proportion (86%; n = 160) had completed at least senior secondary education. With respect to gender identity, sexual behavior, and sexual orientation: 81.6% (n = 151) identified as a man, 2.2% (n = 4) identified as a woman, 15.7% (n = 29) identified as both man and woman and 0.5% (n = 1) identified as neither man or woman; 49.7% (n = 89) reported sex with only men; 50.5% (n = 94) identified has gay, 40.5% (n = 75) identified as bisexual, 4.3% (n = 8) identified as straight, 1.1% (n = 2) were not sure about their sexual orientation, and 3.8% (n = 7) did not want to answer. With respect to HIV status, 18.8% (n = 35) reported they were living with HIV, 45.2% (n = 84) reported they were HIV negative, 26.3% (n = 49) reported they did know, and 9.7% (n = 18) did not want to answer.

### Prevalence of symptoms of Common Mental Disorders (CMDs)

Of the 186 participants, 186 contributed PHQ-9 and GAD-7 scores and 168 contributed PC-PTSD-5 scores. With respect to missingness on the PHQ-9, six participants were missing responses to one question and one participant was missing responses to two questions. All PHQ-9 scores for the participants with missing PHQ-9 data were imputed. One participant was missing a response to the ninth question of the PHQ-9. For the GAD-7, 52 participants were missing responses to one question and two participants were missing responses to two questions. All GAD-7 scores for the participants with missing data were imputed. For the PC-PTSD-5, 18 participants were missing responses to all of the PC-PTSD-5 questions and four participants who reported a traumatic event were missing responses to one question. Only PC-PTSD-5 scores for the four participants missing responses to one question were imputed. See S1 Appendix for further details on the assessment of missingness and the impact of imputation.

Among the sample, the prevalence of symptoms of mild to severe depression was 52.1% and the prevalence of symptoms of moderate to severe depression was 20.4% (Table 2). Further, 35.1% of participants reported suicidality on the 9th item of the PHQ-9. The prevalence of probable mild to severe and moderate to severe anxiety were 50.5% and 12.3%, respectively. With respect to PTSD, 62.5% of participants reported ever experiencing a traumatic event and the prevalence of symptoms of PTSD was 31.5%. Psychiatric multimorbidity was common; 50.6% reported at least two probable common mental disorders (CMDs).

The prevalence of symptoms of CMDs and suicidality was higher among those who self-reported they were living with HIV compared to HIV negative: moderate to severe depression 25.7 vs 20.5%; suicidality 45.7% vs 36.1%; moderate to severe anxiety 14.3% vs 9.5%; and PTSD 50.0% vs. 30.8%.

## Discussion

This small cross-sectional study documented a high prevalence of symptoms of CMDs among a non-probability sample of MSM in Ghana. In comparison, only around 10% of Ghana's entire population is estimated to have any mental disorder.

**Table 1. Participant characteristics (N = 186).**

| | Mean (SD), Median [Min, Max] or N (%) |
|---|---|
| Age | 25.2 (4.5), 24.0 [18, 45] |
| **Education** | |
| No formal education | 1 (0.5%) |
| Primary Education | 4 (2.2%) |
| Middle/Junior High School | 21 (11.3%) |
| Senior Secondary | 99 (53.2%) |
| Tertiary | 61 (32.8%) |
| **Residence*** | |
| Accra | 94 (50.8%) |
| Kumasi | 91 (48.2%) |
| **Gender Identity*** | |
| Man | 151 (81.6%) |
| Woman | 4 (2.2%) |
| Both Man and Woman | 29 (15.7%) |
| Neither | 1 (0.5%) |
| **Sexual Behavior (Who do you have sex with?)\*\*** | |
| Men only | 89 (49.4%) |
| Both Men/Women | 91(50.6%) |
| **Sexual Orientation** | |
| Gay | 94 (50.5%) |
| Straight | 8 (4.3%) |
| Bisexual | 75 (40.3%) |
| Not Sure | 2 (1.1%) |
| Don't want to answer | 7 (3.8%) |
| **HIV Status** | |
| HIV Positive | 35 (18.8%) |
| HIV Negative | 84 (45.2%) |
| I don't know | 49 (26.3%) |
| I don't want to answer | 18 (9.7%) |

*missing n=1;

**missing n= 5

[18] Our findings corroborate the limited research from sub-Saharan Africa, and fill a gap in available data from West Africa. For example, in South Africa, MSM similarly reported a high prevalence of depression (44%) and suicidality (56%); in Lesotho 16% of MSM reported depression; in Tanzania, 46.% of MSM reported elevated depressive symptoms; and in Kenya, 53.2% of sexual and gender minority people reported PTSD and 26.1% depression [6,19–21]. These astoundingly high reports of depression, suicidality, anxiety, experienced trauma and PTSD demonstrate a critical unmet need for mental health care. Psychosocial interventions have shown efficacy at improving mental health among MSM [22]. Sexual health programming for MSM presents a clear opportunity for integration of mental health services to address the complex needs of this vulnerable population [1].

Our findings also support research suggesting a higher prevalence of CMDs – particularly depression – among MSM who are living with HIV [23,24]. Many factors may contribute to elevated risk of CMDs among MSM living with HIV including access to MSM-friendly HIV treatment, HIV-related stigma, isolation, and exclusion – even from within the MSM community, social support,

**Table 2. Prevalence of symptoms of depression, anxiety and PSTD, by HIV status.**

| N (%, 95% CIs) | Total | HIV Status | | |
|---|---|---|---|---|
| | | **Negative** | **Positive** | **Don't Know** |
| | | **(n = 84)** | **(n = 35)** | **(n = 49)** |
| **Depression (n = 186)** | | | | |
| Mild (PHQ-9 5–9) | 59 (31.7, 25-38.4%) | 32 (38.1, 27.7-48.5%) | 10 (28.6, 13.6-43.5%) | 14 (28.6, 15.9-41.2%) |
| Moderate (PHQ-9 10–14) | 27 (14.5, 9.5-19.6%) | 12 (14.3, 6.8-21.8%) | 7 (20.0, 6.7-33.3%) | 6 (12.2, 3.1-21.4%) |
| Severe (PHQ-9 ≥ 15) | 11 (5.9, 2.5-9.3%) | 5 (6.0, 0.9-11%) | 2 (5.7, -2.0-13.4%) | 2 (4.1, -1.5-9.6%) |
| **Suicidality (Q9 +) (n = 185)** | 65 (35.1, 28.3-42%) | 30 (35.7, 25.5-46%) | 16 (45.7, 29.2-62.2%) | 13 (26.5, 14.2-38.9%) |
| **Anxiety (n = 186)** | | | | |
| Mild (GAD-7 5–9) | 71 (38.2, 31.2-45.2%) | 42 (50, 39.3-60.7%) | 13 (37.1, 21.1-53.2%) | 11 (22.4, 10.8-34.1%) |
| Moderate (GAD-7 10–14) | 20 (10.8, 6.3-15.2%) | 6 (7.1, 1.6-12.7%) | 5 (14.3, 2.7-25.9%) | 7 (14.3, 4.5-24.1%) |
| Server (GAD-7 ≥ 15) | 3 (1.6, -0.2-3.4%) | 2 (2.4, -0.9-5.6%) | 0 | 1 (2.0, -1.9-6%) |
| **PTSD (n = 168)** | | | | |
| Experienced traumatic event | 105 (62.5, 55.2-69.8%) | 48 (61.5, 50.7-72.3%) | 24 (80, 65.7-94.3%) | 21 (50.0, 34.9-65.1%) |
| PTSD (PC-PTSD-5 ≥ 4) | 53 (31.5, 24.5-38.6%) | 24 (30.8, 20.5-41%) | 15 (50, 32.1-67.9%) | 9 (21.4, 9-33.8%) |
| **Number of CMDs (n = 168)** | | | | |
| 1 | 19 (11.3, 6.5-16.1%) | 9 (11.5, 4.4-18.6%) | 5 (16.7, 3.3-30%) | 3 (7.1, -0.6-14.9%) |
| 2 | 54 (32.1, 25.1-39.2%) | 27 (34.6, 24.1-45.2%) | 8 (26.7, 10.8-42.5%) | 12 (28.6, 14.9-42.2%) |
| 3 | 34 (20.2, 14.2-26.3%) | 17 (21.8, 12.6-31%) | 9 (30, 13.6-46.4%) | 5 (11.9, 2.1-21.7%) |

CMD = Common Mental Disorders, inclusive of mild to severe depression, mild to severe anxiety, and PTSD

opportunistic infections, unemployment, stress and poverty [21,25–28]. However, very limited research unpacking the mechanisms underlying CMDs among MSM living with HIV in sub-Saharan Africa – particularly within West Africa – has been conducted [23]. While our study was unfortunately neither powered nor designed to explore factors driving differences in CMDs by HIV status, further investigation of risk and protective factors for mental health broadly – inclusive of unhealthy substance use – is urgently needed.

## Limitations

The study employed non-diagnostic mental health screening measures to assess prevalence at standard cutoff scores. While the PHQ-9 and GAD-7 have been used in Ghana [29–31], the screeners have yet to be validated against a diagnostic gold-standard in Ghana. While it is possible that imputation of missing data, particularly on the GAD-7, may have biased the prevalence estimates, the expected impact of imputation on total scores is minimal (see S1 Appendix). This analysis includes a unique population of MSM – participants from an intervention trial. The mental health survey participants had recently participated in a stigma-reduction and HIV prevention intervention that targeted intra- and interpersonal intersectional stigma and encouraged peer-support. Given the protective nature of stigma-reduction and peer-support interventions, our findings likely underestimate the prevalence of symptoms of CMDs among MSM in Ghana. The sample size is small, which limits our ability to assess differences among sub-groups of MSM. While we were fortunate to be able to leverage the close-out of the RCT to measure the prevalence of mental health concerns, we are limited in our ability to investigate factors associated with poor mental health. However, the data presented provides useful insights into the magnitude of mental health concerns among MSM both with and without HIV.

## Conclusion

This descriptive short report presents valuable mental health prevalence data among a particularly vulnerable population, thus helping to fill a crucial gap in the literature, particularly in West Africa. Further research is urgently needed to

understand risk and protective factors for CMDs among MSM in Ghana and elsewhere in sub-Saharan Africa. These findings also signal the need for investments in psychosocial support for MSM beyond HIV prevention and treatment as well as in expanding access to services for mental health promotion and treatment. Recognizing the reinforcing syndemics of HIV and mental health problems that hamper the wellbeing of MSM, culturally and contextually appropriate interventions that integrate in HIV and mental health care for MSM will be critical to addressing their healthcare needs [1].

## Supporting information

**S1 Appendix.  Assessment of missingness and impact of imputation.**
(DOCX)

**S1 Checklist.  Inclusivity checklist.**
(DOCX)

## Acknowledgments

We are grateful for the generosity of the study participants and the expertise of our partner organizations and their staff: Priorities on Rights and Sexual Health (PORSH) in Accra, Youth Alliance for Health and Human Rights (YAHR), in Kumasi, and Educational Assessment Research Centre (EARC), in Accra.

## Author contributions

**Conceptualization:** Melissa A. Stockton, Kwasi Torpey, Laura Nyblade, LaRon E Nelson.

**Formal analysis:** Melissa A. Stockton, Sangchoon Jeon.

**Funding acquisition:** Kwasi Torpey, Laura Nyblade, LaRon E Nelson.

**Methodology:** Melissa A. Stockton.

**Project administration:** Samuel Akyirem, Edem Zigah, Nii Dromo Wallace-Atiapah, Gamji Rabiu Abu-Ba'are, Richard Panix Amo-Oto, Irene Ofori, Michael Adu.

**Supervision:** Samuel Akyirem, Edem Zigah, Nii Dromo Wallace-Atiapah, Gamji Rabiu Abu-Ba'are, Richard Panix Amo-Oto.

**Writing – original draft:** Melissa A. Stockton.

**Writing – review & editing:** Samuel Akyirem, Edem Zigah, Nii Dromo Wallace-Atiapah, Gamji Rabiu Abu-Ba'are, Richard Panix Amo-Oto, Irene Ofori, Michael Adu, Kwasi Torpey, Laura Nyblade, LaRon E Nelson.

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
