## [Decision Letter · Decision Letter 0]

PMEN-D-24-00479

Prevalence of Probable Common Mental Disorders and Suicidality among Sexually Diverse Men who Have Sex with Men in Ghana

PLOS Mental Health

Dear Dr. Stockton,

Thank you for submitting your manuscript to PLOS Mental Health. After careful consideration, we feel that it has merit but does not fully meet PLOS Mental Health’s publication criteria as it currently stands. Therefore, we invite you to submit a revised version of the manuscript that addresses the points raised during the review process.

We look forward to receiving your revised manuscript.

Kind regards,

Marc Eric Santos Reyes

Academic Editor

PLOS Mental Health

Journal Requirements:

 1. Please include a complete copy of PLOS’ questionnaire on inclusivity in global research in your revised manuscript. Our policy for research in this area aims to improve transparency in the reporting of research performed outside of researchers’ own country or community. The policy applies to researchers who have travelled to a different country to conduct research, research with Indigenous populations or their lands, and research on cultural artefacts. The questionnaire can also be requested at the journal’s discretion for any other submissions, even if these conditions are not met.  Please find more information on the policy and a link to download a blank copy of the questionnaire here: https://journals.plos.org/plosmentalhealth/s/best-practices-in-research-reporting. Please upload a completed version of your questionnaire as Supporting Information when you resubmit your manuscript.”  2. We noticed you have some minor occurrence of overlapping text with the following previous publication(s), which needs to be addressed: - https://doi.org/10.1016/j.jad.2024.06.071-
https://doi.org/10.1371/journal.pone.0259324 In your revision ensure you cite all your sources (including your own works), and quote or rephrase any duplicated text outside the methods section. Further consideration is dependent on these concerns being addressed.  3. Please amend your detailed Financial Disclosure statement. This is published with the article. It must therefore be completed in full sentences and contain the exact wording you wish to be published. i. Please clarify all sources of funding (financial or material support) for your study. List the grants (with grant number) or organizations (with url) that supported your study, including funding received from your institution. ii. State the initials, alongside each funding source, of each author to receive each grant.iii. State what role the funders took in the study. If the funders had no role in your study, please state: “The funders had no role in study design, data collection and analysis, decision to publish, or preparation of the manuscript.”iv. If any authors received a salary from any of your funders, please state which authors and which funders.  4. In the online submission form, you indicated that “Deidentified data may be made available upon reasonable request.”.  All PLOS journals now require all data underlying the findings described in their manuscript to be freely available to other researchers, either 1. In a public repository, 2. Within the manuscript itself, or 3. Uploaded as supplementary information. This policy applies to all data except where public deposition would breach compliance with the protocol approved by your research ethics board. If your data cannot be made publicly available for ethical or legal reasons (e.g., public availability would compromise patient privacy), please explain your reasons by return email and your exemption request will be escalated to the editor for approval. Your exemption request will be handled independently and will not hold up the peer review process, but will need to be resolved should your manuscript be accepted for publication. One of the Editorial team will then be in touch if there are any issues.

Additional Editor Comments (if provided):

Reviewers' comments:

Reviewer's Responses to Questions

**Comments to the Author**

1. Does this manuscript meet PLOS Mental Health’s publication criteria ? Is the manuscript technically sound, and do the data support the conclusions? The manuscript must describe methodologically and ethically rigorous research with conclusions that are appropriately drawn based on the data presented.

Reviewer #1: Partly

Reviewer #2: Yes

Reviewer #3: Yes

2. Has the statistical analysis been performed appropriately and rigorously?

Reviewer #1: Yes

Reviewer #2: Yes

Reviewer #3: Yes

3. Have the authors made all data underlying the findings in their manuscript fully available (please refer to the Data Availability Statement at the start of the manuscript PDF file)?

Reviewer #1: No

Reviewer #2: Yes

Reviewer #3: No

4. Is the manuscript presented in an intelligible fashion and written in standard English?

Reviewer #1: Yes

Reviewer #2: Yes

Reviewer #3: Yes

5. Review Comments to the Author

Reviewer #1: Technical Soundness and Data Support

The study uses validated screening tools (PHQ-9, GAD-7, PC-PTSD-5) for assessing mental health conditions. These tools are widely accepted, though not validated specifically in Ghana, which is acknowledged as a limitation.

Ethical approval was obtained from multiple institutional review boards, and participants gave informed consent.

The methods and participant eligibility criteria are clear and appear appropriate for the research aims.

The conclusions are appropriately aligned with the data presented. They emphasise the high prevalence of mental health disorders and suicidality among MSM in Ghana and highlight the need for targeted mental health interventions.

The statistical methods (univariate analyses) are appropriate for a prevalence study.

Missing data was addressed using a Monte Carlo Markov Chain imputation method, which is an advanced and suitable approach.

The manuscript states that de-identified data may be made available upon reasonable request. However, for full compliance with PLOS policies, the data should ideally be fully accessible without restriction. Clarification may be needed.

The manuscript is generally intelligible and written in standard English. However, there are minor typographical errors (e.g., "Questionaire" instead of "Questionnaire") and areas where clarity could be improved, such as sentence structure in some sections of the methods and discussion.

The manuscript adheres to academic conventions but may require slight adjustments to meet PLOS formatting guidelines (e.g., headings, reference formatting).

Typographical/Grammatical Errors:

- Replace “Questionaire” with “Questionnaire.”

- Inconsistent spacing between words (e.g., "DSMsymptoms" instead of "DSM symptoms") should be corrected.

- Consider rephrasing for clarity in sentences like: *"This analysis includes a unique population of MSM; the mental health survey participants had recently participated in a stigma-reduction and HIV-prevention intervention RCT that aimed at addressing intra-and inter-personal levels of intersectional stigma and ultimately encouraged peer-support."

Recommendations for Authors

1. Address the availability of data by ensuring unrestricted access to comply fully with PLOS policies.

2. Conduct a thorough proofread or engage a professional editor to address minor typographical and grammatical issues.

3. Consider rephrasing and streamlining dense sentences for enhanced readability.

Reviewer #2: The authors present an interesting study overall. However, in the abstract, possibly some more elaboration in terms of the background could be included, apart from simply prevalence and a general description of the country's situation. Despite this being a short report, a few details could be added for depth and breadth. The introduction contains such information, but the abstract may pull in more of these to improve it.

A statement about the main purpose of the study may be explicitly stated, including its hypothesis, to establish the aim of the study, as consistent with the succeeding parts.

Perhaps some statements about the validation of the tools used from the English language to local ones would have been a point of improvement.

Apart from the above mentioned points, the short article appears to be a great contribution especially using the data coming from a bigger project.

Reviewer #3: Dear authors,

I appreciate the opportunity to review your manuscript titled "Prevalence of Probable Common Mental Disorders and Suicidality among Sexually Diverse Men Who Have Sex with Men in Ghana." Your study addresses a highly relevant public health issue, providing valuable evidence on the high prevalence of mental health disorders in a particularly vulnerable and underrepresented population. The syndemic approach and the exploration of the intersection between mental health, HIV, and stigma are essential for understanding these dynamics in the Ghanaian context. Below, I provide a series of comments and suggestions to strengthen the manuscript by improving methodological clarity, enhancing the depth of analysis, and further discussing the findings of existing literature.

-In the title, the phrase "Prevalence of Probable Common Mental Disorders" may create ambiguity, as it is not entirely clear whether "probable" refers to diagnoses inferred from screening tools or to a broader methodological assumption. If the study relies solely on screening instruments (PHQ-9, GAD-7, PC-PTSD-5) rather than clinical diagnoses confirmed by mental health professionals, it would be more precise to explicitly indicate that the reported prevalence is based on screening results rather than definitive diagnoses. Using the term "Screening-Based Prevalence" could help clarify this distinction.

- Although the PHQ-9, GAD-7, and PC-PTSD-5 are widely used internationally, they have not been validated in the Ghanaian context. The scores may not accurately reflect symptomatology in this population without proper cultural validation. It is recommended that the authors discuss this limitation in greater depth and, if possible, include information on the semantic and conceptual equivalence of the scales used.

- Additionally, reliability metrics (alpha or omega) for the instruments used in the study sample are not reported. Without this information, it is difficult to assess the internal consistency of the scales in this specific context. Including these indicators is recommended to strengthen the validity of the results.

- Although the authors mention imputation using the Monte Carlo Markov Chain (MCMC) method, there is no analysis of the type of missing data (MCAR, MAR, or MNAR). Identifying whether the data is completely random (MCAR), random but conditioned on observed variables (MAR), or not random (MNAR) is crucial to assessing the adequacy of the imputation. Including an exploratory analysis of missing data patterns is recommended before justifying the use of MCMC. Additionally, a comparison between imputed values and original data could be considered to assess the robustness of the procedure.

- The reported prevalence estimates lack confidence intervals (CIs), which prevents evaluating the precision of the results and their possible variability. Given that the sample comes from a specific population—participants in a clinical intervention trial—and not from a representative sampling, the findings cannot be reliably extrapolated to the general MSM population in Ghana. Including 95% CIs would help reflect the uncertainty in the estimates and reinforce the discussion on the limitations of generalizing the results.

- While studies in other African countries are mentioned, the comparison is superficial. The authors could strengthen the discussion by contrasting the findings with studies that have used similar methodologies in comparable sociocultural contexts.

- A higher prevalence of mental health problems is reported among MSM living with HIV, but the underlying mechanisms that could explain this association are not explored. Factors such as access to treatment, internalized stigma, and social support may play a key role in this relationship. A more detailed discussion of these aspects would enrich the analysis.

6. PLOS authors have the option to publish the peer review history of their article (what does this mean? ). If published, this will include your full peer review and any attached files.

**Do you want your identity to be public for this peer review?** For information about this choice, including consent withdrawal, please see our Privacy Policy .

Reviewer #1: **Yes: ** Simon Browes

Reviewer #2: **Yes: ** Peejay D. Bengwasan

Reviewer #3: No

---

## [Decision Letter · Decision Letter 1]

Symptoms of Common Mental Disorders and Suicidality among Sexually Diverse Men who Have Sex with Men in Ghana

PMEN-D-24-00479R1

Dear Ms. Stockton,

We are pleased to inform you that your manuscript 'Symptoms of Common Mental Disorders and Suicidality among Sexually Diverse Men who Have Sex with Men in Ghana' has been provisionally accepted for publication in PLOS Mental Health.

Best regards,

Karli Montague-Cardoso

Executive Editor

PLOS Mental Health

Reviewer Comments (if any, and for reference):

Reviewer's Responses to Questions

**Comments to the Author**

1. If the authors have adequately addressed your comments raised in a previous round of review and you feel that this manuscript is now acceptable for publication, you may indicate that here to bypass the “Comments to the Author” section, enter your conflict of interest statement in the “Confidential to Editor” section, and submit your "Accept" recommendation.

Reviewer #1: All comments have been addressed

Reviewer #2: All comments have been addressed

2. Does this manuscript meet PLOS Mental Health’s publication criteria ? Is the manuscript technically sound, and do the data support the conclusions? The manuscript must describe methodologically and ethically rigorous research with conclusions that are appropriately drawn based on the data presented.

Reviewer #1: Yes

Reviewer #2: (No Response)

3. Has the statistical analysis been performed appropriately and rigorously?

Reviewer #1: Yes

Reviewer #2: Yes

4. Have the authors made all data underlying the findings in their manuscript fully available (please refer to the Data Availability Statement at the start of the manuscript PDF file)?

Reviewer #1: No

Reviewer #2: Yes

5. Is the manuscript presented in an intelligible fashion and written in standard English?

Reviewer #1: Yes

Reviewer #2: Yes

6. Review Comments to the Author

Reviewer #1: Thank you for resubmitting the revised manuscript.

The manuscript is technically sound, methodologically robust, and ethically conducted. It appropriately uses validated screening tools (PHQ-9, GAD-7, PC-PTSD-5) to assess symptoms of depression, anxiety, PTSD, and suicidality in a hard-to-reach population.

Though the instruments are not validated in Ghana, the authors:

Acknowledge this limitation clearly

Reference existing studies showing good psychometric performance in Ghana and Côte d’Ivoire

Provide internal reliability metrics (α > 0.8) for this sample

The authors have improved the description of imputation methods (Monte Carlo Markov Chain), and provided a thorough exploration of the impact of this with clear supplementary analyses.

The conclusions are appropriate and supported by the data.

The manuscript stops short of causal interpretation, emphasising instead the urgent need for mental health services, especially for MSM living with HIV.

Statistical analysis is appropriate for a prevalence study:

Univariate analysis is used to estimate the proportion of CMDs and suicidality.

Proportions are presented with 95% confidence intervals, as requested by reviewers.

Missing data were addressed with an MCMC imputation method, and the robustness of results was tested with pre-imputation, complete case, and partial case comparisons.

This approach was explained clearly in the revised methods and limitations, and detailed results were added in the S1 Appendix.

The authors have clarified that ethics board restrictions prevent public data sharing. Instead, de-identified data may be available upon reasonable request with IRB approval.

This is a reasonable exemption under PLOS’s policy, but editorial approval is still required.

The authors have:

Clearly named the ethics boards involved

Provided appropriate justification

Updated the data availability statement accordingly

The revised manuscript shows significant improvements in clarity, grammar, and typographical accuracy.

The authors:

Fixed terms like “Questionaire” to “Questionnaire”

Smoothed long, dense sentences

Removed redundancy and restructured the abstract and methods for clarity

Removed unclear phrasing such as “DSMsymptoms” and replaced it appropriately

The writing is now coherent, professional, and accessible. Minor style inconsistencies (e.g., “Server” instead of “Severe” in one table row) could still be corrected at proof stage.

Reviewer #2: As the authors have addressed my previous comments, I have no suggestions to add thus far.

Thank you and congratulations!

7. PLOS authors have the option to publish the peer review history of their article (what does this mean? ). If published, this will include your full peer review and any attached files.

**Do you want your identity to be public for this peer review?** For information about this choice, including consent withdrawal, please see our Privacy Policy .

Reviewer #1: **Yes: ** Simon Browes

Reviewer #2: **Yes: ** Peejay D. Bengwasan
